# Merkel Cell Polyoma Virus and Cutaneous Human Papillomavirus Types in Skin Cancers: Optimal Detection Assays, Pathogenic Mechanisms, and Therapeutic Vaccination

**DOI:** 10.3390/pathogens11040479

**Published:** 2022-04-16

**Authors:** Ramona Gabriela Ursu, Costin Damian, Elena Porumb-Andrese, Nicolae Ghetu, Roxana Gabriela Cobzaru, Catalina Lunca, Carmen Ripa, Diana Costin, Igor Jelihovschi, Florin Dumitru Petrariu, Luminita Smaranda Iancu

**Affiliations:** 1Department of Preventive Medicine and Interdisciplinarity (IX)—Microbiology, “Grigore T. Popa” University of Medicine and Pharmacy, 700115 Iasi, Romania; ramona.ursu@umfiasi.ro (R.G.U.); costin.damian@d.umfiasi.ro (C.D.); cobzaru.roxana@umfiasi.ro (R.G.C.); catalina.lunca@umfiasi.ro (C.L.); ripa.carmen@umfiasi.ro (C.R.); diana.costin@umfiasi.ro (D.C.); jelihovschi.igor@umfiasi.ro (I.J.); florin.petrariu@umfiasi.ro (F.D.P.); luminita.iancu@umfiasi.ro (L.S.I.); 2Department of Medical Specialties (III)—Dermatology, “Grigore T. Popa” University of Medicine and Pharmacy, 700115 Iasi, Romania; elena.andrese1@umfiasi.ro; 3Department of Plastic Surgery, Regional Oncology Institute, 700483 Iasi, Romania

**Keywords:** skin cancer, Merkel cell polyomavirus, polyomaviruses, beta HPV types, targeted therapy

## Abstract

Oncogenic viruses are recognized to be involved in some cancers, based on very well-established criteria of carcinogenicity. For cervical cancer and liver cancer, the responsible viruses are well-known (e.g., HPV, HBV); in the case of skin cancer, there are still many studies which are trying to identify the possible viral etiologic agents as principal co-factors in the oncogenic process. We analysed scientific literature published in the last 5 years regarding mechanisms of carcinogenicity, methods of detection, available targeted therapy, and vaccination for Merkel cell polyomavirus, and beta human papillomavirus types, in relation to skin cancer. This review is targeted at presenting the recent findings which support the involvement of these viruses in the development of some types of skin cancers. In order to optimize the management of skin cancer, a health condition of very high importance, it would be ideal that the screening of skin cancer for these two analysed viruses (MCPyV and beta HPV types) to be implemented in each region’s/country’s cancer centres’ molecular detection diagnostic platforms, with multiplex viral capability, optimal sensitivity, and specificity; clinically validated, and if possible, at acceptable costs. For confirmatory diagnosis of skin cancer, another method should be used, with a different principle, such as immunohistochemistry, with specific antibodies for each virus.

## 1. Introduction

The International Agency for Research on Cancer (IARC) recognized the Epstein-Barr virus (EBV), hepatitis B virus (HBV), hepatis C virus (HCV), Kaposi’s sarcoma herpes virus (or human gammaherpesvirus 8), human immunodeficiency virus 1 (HIV-1), several human papillomavirus (HPV) types, and human-T lymphotropic retrovirus-1 (HTLV-1) as biological agents involved in human carcinogenesis. Criteria used to prove the involvement of these viruses in the tumorigenesis process was based on analyzing exposure data, studies on cancer in humans and in animal models, and identification of relevant data providing mechanistic insight. Exposure data refer to general information about the agent, analysis and detection methods regarding sensitivity and specificity, occurrence, and exposure. Research of cancer in humans analyzed the type of studies (cohort studies, case-control studies), meta-analyses and pooled analyses, temporal effect, the use of biomarkers in epidemiological studies, and criteria of causality. Model animal studies investigated the qualitative and quantitative aspects, and mechanistic insight referred to toxicokinetic data, mechanisms of carcinogenesis which identified functional changes at the cellular level and alterations at the molecular level [1].

The estimated age-standardized incidence rates in 2020 for skin melanoma and non-melanoma skin cancer, both genders, all ages, WHO Europe, mention Switzerland, Ireland, and The Netherlands in the first three places, with age-standardized rates (ASR) 71.1, 63.3, and 61.5, respectively, with Romania having an ASR of 12.0 [2]. This data collection regarding the incidence of cancer is powered by population-based cancer registries which are available in developed countries; this led to the idea that in countries without population-based cancer registries, the incidence of cancers is underreported. Skin cancer is known to have several risk factors, such as sun exposure, BRAF mutation in melanoma patients, and some molecular factors [3,4]. The higher incidence of NMSC (nonmelanoma skin cancer) in immunocompromised patients points to a possible viral origin [5]. In this review, we proposed to analyze the recent findings regarding involvement of two viruses in skin cancers: Merkel cell polyomavirus (MCV or MCPyV), and beta human papillomavirus (HPV) types.

### 1.1. Merkel Cell Polyomavirus 

The *Polyomaviridae* family includes numerous small, icosahedral, non-enveloped viruses, which have a double-stranded DNA genome that is approximately 5000 base pairs in length, and it is packed together with histones uptake from the host cells. These viruses have a wide range of hosts, including mammals, birds, and fish [6]. The International Committee on Taxonomy of Viruses (ICTV) currently recognizes eight different genera of polyomaviruses (*Alpha-*, *Beta-*, *Gamma*, *Delta-*, *Epsilon-*, *Zeta-*, *Eta-* and *Thetapolyomavirus*), comprising a total of 117 species [7]. The genetic diversity of these viruses is also very great, and a characteristic co-speciation with their hosts has been observed, which is a result of genetic recombination, as it has been observed for papillomaviruses [8]. Many of the viruses in this family are associated with an oncogenic capacity in animal hosts, which has been observed since the discovery of murine polyomavirus in the 1950s. The *Polyomaviridae* family was given this name because of the numerous types of tumors they can induce (polyoma) [9]. 

Of the human polyomaviruses, MCPyV was the first one for which evidence of carcinogenic potential has been observed, in a rare and aggressive form of skin cancer named Merkel cell carcinoma (MCC). MCPyV was first discovered at the Pittsburgh Cancer Institute in 2008, using digital transcriptome subtraction assays. The authors detected that the viral DNA integrated within the tumoral cells’ genome in a clonal pattern, in 6/8 MCPyV-positive MCCs, suggesting that MCPyV infection and integration preceded clonal expansion of the tumoral cells. MCPyV was then first considered to have a contributing factor in the pathogenesis of MCC [10]. Four years later, scientists from 11 countries met at IARC, to evaluate the carcinogenicity of MCPyV, and their research has been published in a monograph and in a *Lancet* paper. By analyzing all the research studies published since its discovery, the authors concluded that there is powerful mechanistic evidence that MCPyV can directly contribute to the development of a large proportion of MCCs. Using PCR, many independent laboratories have detected MCPyV DNA in about three quarters of more than 1000 MCC cases [11,12]. 

In 2017–2018, a multidisciplinary team from important research centers of many continents (e.g., German Cancer Research Centre, DKFZ, Heidelberg, Germany, Department of Melanoma Medical Oncology, Division of Cancer Medicine, MD Anderson Cancer Centre, Houston, TX, USA, Peter McCallum Cancer Centre, Melbourne, Australia, to name just a few) published three reviews regarding MCPyV in relation to skin cancer [13,14,15].

Becker JC et al., classified MCCs in MCPyV positive and negative and mentioned that in the countries with low UV exposure, MCPyV is present in most of the skin cancers, in stark contrast with countries with high UV exposure where the virus is absent in MCCs. It is interesting that both MCC types have similar phenotypes, and several tissue markers were detected in skin cancer that may be positive or not for MCPyV, including apoptosis regulator B-cell lymphoma-2 (BCL2), cytokeratin 20, neural cell adhesion molecule 1, CD99, CD99 antigen, epithelial cell adhesion molecule, huntingtin-interacting protein 1, neuron-specific enolase, and neurogenic locus notch homologue protein 1. The MCPyV-specific MCC viral markers are large T antigen and small T antigen. The authors mentioned that, in the case of viral-positive MCC cases, the genetic aberrations observed are from perturbations of signaling pathways by antigens and genome integration; meanwhile, in the case of UV exposure, other alterations were detected, such as deletions, translocations, and point mutations [13].

In the second review [14], members of the EU IMMOMEC (European Union Immune Modulating strategies for treatment of Merkel Cell Carcinoma) presented the actual available therapy that is efficient for this type of skin cancer: the immune checkpoint-inhibiting antibodies pembrolizumab and avelumab [specifically, the programmed death protein 1 (PD-1) and programmed death-ligand 1 (PD-L1) blocking antibodies]. This new therapy seems to be efficient in more than half of the treated MCC patients. Still, a targeted therapy is still necessary, as many MCC patients are immunosuppressed and their response to immune checkpoint inhibition is not possible [15].

In 2018, the International Workshop on Merkel Cell Carcinoma Research (IWMCC) working group underlined some open research questions regarding this primary cutaneous neuroendocrine carcinoma, MCC: the multidisciplinary research team (e.g., virology, pathology, oncology, dermatology) raised awareness regarding future targeted therapy in both MCPyV positive and MCPyV negative cases of MCC, and about the optimal detection assay for this virus [15]. 

### 1.2. Cutaneous HPV Types 

The *Papillomaviridae* family is comprised of small, icosahedral, non-enveloped viruses with a double-stranded DNA genome, and are also characterized by a great genetic diversity and wide range of hosts, including mammals, birds, reptiles, and fish. They also have a known oncogenic potential in humans, most importantly in the development of cervical cancer, but also vulvar, vaginal, penile, and oropharyngeal cancers. The human papillomaviruses which are associated with those cancers are also called mucosal, high-risk, or alpha HPV types [16].

The first classification of cutaneous papillomaviruses was performed by de Villiers EM et al., in 2004 [17]. In 2012, the IARC monograph reported that, up to that moment, there was no HPV type which could be considered to cause skin cancer, due to the lack of consistency of the published data. In 2012, it was considered that the role of HPV types in skin cancers could be complex, possibly associated with other co-factors, such as UV exposure [1]. In 2013, over 170 human papillomavirus types were reported to be associated with different clinical manifestations in humans, with the skin being the main site, followed by mucosa (vagina, mouth) and gut [18].

In a similar manner to MCPyV, for cutaneous HPV types, important reviews were recently published, with a different view regarding the involvement of these viruses in skin cancer.

Venuti A et al., analyzed the “cross-talk” between cutaneous HPV types and the immune system, in a journal of the Royal Society; they mentioned the “hit-and-run” hypothesis, having the ability to initiate the first steps of UV-driven skin carcinogenesis, a different mechanism of carcinogenesis, in comparison with that of mucosal HPV types responsible for cervical cancer. The authors underlined the necessity of understanding the cross-talk with host cell-autonomous and extrinsic immunity for it to be possible to identify novel therapies against beta HPV, besides their sensitivity to interferon regulatory factors [19].

Gheit T., 2019, an IARC researcher with impressive experience in HPV testing and analyzing, presented the main features and functions of the early and late gene products from alpha and beta HPV types. Interestingly, for E6 and E7 genes known as oncogenic in cervical cancer, different functions are underlined: both genes are not required for the maintenance of the cancer phenotype. E6 interacts with the Notch pathway and promotes the transformation process of the infected keratinocytes, and inhibits the differentiation of HPV8-expressing keratinocytes by targeting the PDZ domain-containing protein syntenin 2. E7 from HPV38 shows the ability to counteract p53-mediated apoptosis by inducing an accumulation of the p73 isoform, 1Np73 [16]. The author was in support of the hypothetical carcinogenesis mechanism of the previous review [19], mentioning that E6 and E7 expression appear to be required only at the initial step of skin carcinogenesis by exacerbating the deleterious effects of UV radiation [16].

Rollison DE et al., 2019, mentioned that since the first meeting group at IARC regarding beta HPV types, 50 types of cutaneous HPV have been identified from a total of 200 HPV types. The authors underlined the importance of UV as a co-factor in skin carcinogenesis, in the case of constant stress, and it is considered that cutaneous HPV types facilitate DNA damage accumulation induced by UV radiation. This review used high-risk HPV (HR-HPV) types as a comparison, and the authors are confident that, if for cervical cancer developing three vaccines (bi, tetra, and nonavalent) was possible, it will be feasible to create a vaccine against beta HPV types [20].

Given these recent data regarding skin cancer and the association with two potentially oncogenic viruses, we aim in this review to present updates regarding detection methods, carcinogenetic mechanisms, and the availability of therapeutic vaccination for MCPyV and cutaneous HPV types.

### 1.3. MCV DNA Detection Assays 

We analyzed the articles published in the last 5 years using the following keywords:”skin cancer Merkel cell polyomavirus detection assay”. From all 34 studies suitable for our research, only 13 studies were selected. Since 2007, the detection methods for Merkel cell polyomavirus have evolved from serological diagnosis MCV-oncoprotein antibody detection [21], to testing by simultaneous complementary molecular techniques (classical and qPCR) [22,23,24,25,26,27], or by double analyzing MCC with molecular and immunohistochemical analyses [28]. This double testing underlines the necessity of confirmatory diagnosis by two different or complementary assays. An interesting study performed in France analyzed the circulating tumoral cells (CTCs) in blood samples for patients with MCC, and the authors remarked on the tumor heterogeneity [29]. A very comprehensive study was the one realized at Bethesda, MD USA, in 2020, in which the authors used deep sequencing with OncoPanel, a clinically implemented, next-generation sequencing assay targeting over 400 cancer-associated genes; they observed the value of high-confidence virus detection for identifying molecular mechanisms of UV and viral oncogenesis in MCC [30]. MCV DNA was detected in formalin-fixed and paraffin-embedded (FPPE) MCC samples, with different prevalence, from 10% up to 90%. The most common primers used were targeting the sT gene, VP1 and NCCR regions of the genome, and large T-antigen (LTAg) gene; the sequence of used primes are presented by the authors [21,22,23,24,25,26,27,28,29,30,31,32,33] (Table 1).

Oncogenic transformation by MCPyV is hypothesized to require two events: the integration of the viral genome into the host genome, and the truncation of large Tantigen to render the viral genome incapable of replication. In the viral positive MCCs, the small T antigen has an important role in carcinogenesis: it is known to transform rat-1 fibroblasts in culture. Research carried out on transgenic mouse models has shown that the expression of small T antigen was transformative in various organ systems, including in the epidermis [15].

The carcinogenetic mechanisms identified from 2017 to 2022 were determined in different modalities, beginning with evaluating the MCPyV cultivation on cell lines, to assessing the expression of specific genes. The outcomes of the analyzed studies were correlated with possible future targeted therapies, including for metastatic MCC, and even with future vaccinations [34,35,36,37,38,39,40] (Table 2).

Skin biopsies from different skin cancer types were analyzed for the study of cutaneous HPV types, alone or in parallel, with samples from healthy skin as a comparison. The authors have used only molecular biology techniques, beginning with classical PCR, followed by hybridization, qPCR, multiplex genotyping, and NGS. The primers used were targeting the E1 β-HPV gene fragment, two pairs of general degenerate primers CP65–70 (CP65/70 and CP66/69, consensus primer pair FAP (FAP59\FAP64) targeting the 5′end of the L1 ORF, FAP and PGMY-GP + primer systems, and E7 gene for HPV types. The analyzed studies found HPV types in different percentages in skin cancer and newly identified HPV types were reported. The authors underline the need of optimizing the sensitivities of the used assays and the necessity of confirmatory methods [41,42,43,44,45] (Table 3).

The studies that focus on the carcinogenesis of cutaneous HPV types in skin cancer range from identifying these viruses as co-factors, observing mutations in infected mice, to studying the transforming activity of beta HPV types [46,47,48] (Table 4).

## 2. Discussion and Conclusions

In this review, we analyzed the studies published in the last 5 years regarding detection methods and carcinogenesis mechanisms for two viruses associated with skin cancer: MCPyV and cutaneous HPV types. These above-mentioned assays vary substantially in terms of sensitivity and specificity for the detection of tumour association for both viruses. For MCPyV, double testing was used: molecular and immunohistochemistry for confirmation. For cutaneous HPV types, only PCR and sequencing-based methods were reported. It is obvious that using more sensitive assays, such as NGS, will lead to the detection of more viral-positive tumor cases, in comparison with classical PCR technique alone. The identified carcinogenesis mechanisms were correlated for both viruses, with future targeted therapy and with possible therapeutic vaccination. The clinical utility of detection of the viral-induced tumors is supported by the following papers.

In the multicenter Cancer Immunotherapy Trials Network, phase II trial, more than half (64%) of the MCPyV-positive MCC patients received first-line anti-programmed cell death-1 therapy (Pembrolizumab); the authors reported an improved trend of progression-free survival and overall survival, in comparison with chemotherapy-treated patients [49]. This study was continued, and a more recent paper reported the same efficiency of first-line anti-programmed death-(ligand) 1 (anti-PD-(L)1) therapy in MCC [50].

Another clinical application of MCPyV detection in MCC is the possibility of developing therapeutical vaccinations, as reported by Xu D et al., in 2021. The authors began their research with purification of VP1 capsid protein of MCPyV, and then developed a murine tumor model. The next step consisted in the evaluation of the effects of VP1 therapeutic vaccine, as a result of triple immunization; this new developed vaccine induced strong and durable antitumor effects [51]. Other authors also mentioned the possibility of future development of novel therapies, e.g., cancer vaccines and/or CD4 T-cell therapy, which could provide much-needed adjunctive therapeutic strategies for MCC patients and cancer patients [39].

We did not identify any skin cancer-positive beta HPV type clinical trial, but we noticed a recent study published in Nature (2019), in which the authors used human tissue and animal model studies; they discovered that E7 peptides from β-HPVs activated CD8+ T cells from unaffected human skin. Their findings establish a foundation for immune-based approaches that could block the development of skin cancer by boosting immunity against the commensal HPVs [52]. 

A limitation of our study is the relatively low number of analyzed studies. We performed a systematic search of the PubMed and the EMBASE databases for all the published studies on skin cancer, Merkel cell polyomavirus, and beta human papillomavirus types, using the following search algorithm: skin cancer AND MCPyV/beta HPV types AND detection assays/carcinogenesis mechanism/therapeutic vaccination. We discovered a systematic analysis for studies that was published in English, from the 1st of January 2017 to the 1st of February 2022, which described the methods of detection of these two viruses in skin cancer, their tumorigenesis mechanism in this kind of cancer, and possible therapeutical vaccination approaches. One possible explanation for the relatively low number of studies found is that these viruses have only recently begun to be associated with skin cancer (especially beta HPV types). Another possible explanation is that our research investigation period included the COVID pandemic, which is known to have stopped or delayed patients’ access to medical services, and even that some research groups had delays in their activity.

We identified different molecular assays used for both analyzed viruses (e.g., PCR, real-time PCR, nested PCR, NGS, multiplex PCR), which are expected to have different sensitivities, specificities, and positive/negative predictive values. The studies were performed in just a few countries by established researchers in the field. However, the presence of MCPyV and beta HPV types was not routinely tested for common diagnosis in any country.

To optimize the management of skin cancer, a health condition of very high importance, it would be ideal that the screening of skin cancer for these two analyzed viruses (MCPyV and beta HPV types) be implemented in each region’s/country’s cancer centers’ molecular detection diagnostic platforms, with multiplex viral capability. The diagnostic platform should fulfill the criteria for optimal sensitivity and specificity (as close as possible to 100%), clinically validated (on larger cohorts of testes patients), and if possible, at acceptable costs. This approach could be possible with the apport of health programs, by recognizing the necessity of screening for this possibly viral-induced cancer. One possible model to be followed in skin cancer screening for oncogenic viruses is the case of HPV and cervical cancer. For HPV screening, very strict criteria have been established in a guideline from 2008, for an HPV DNA test requirement: the candidate test should have a clinical sensitivity of not less than 90%, a clinical specificity of not less than 98%, and a high interlaboratory agreement of at least 92% [53]. These strict criteria were fulfilled over the years; recently in 2021, well-recognized researchers proved that they were able to implement screening for HPV with big molecular platforms (e.g., COBAS 6800), with an overall sensitivity of 99.1% and a relative specificity of 99.1% [54]. 

For this scenario to be possible in the case of skin cancer, there is a need for more studies to confirm the etiological link between MCPyV and beta HPV types and skin cancer. Both viruses have DNA genomes; thus, it could be possible to develop a molecular platform for multiplex genotyping assays, and even quantification of these viruses. For confirmatory diagnosis of skin cancer, another method should be used, with a different principle, such as immunohistochemistry, with specific antibodies for each virus. In order for the above directions to be possible, there is a need for more studies to demonstrate the association between skin cancer and these two viruses, and of course, basic research studies to confirm the already described carcinogenesis mechanism.

In conclusion, this review underlines the necessity of interdisciplinary collaboration in assessing skin cancers to understand the natural history of MCPyV and beta HPV types, and to correlate their carcinogenesis mechanisms with future targeted therapies and vaccinations.

## Figures and Tables

**Table 1 pathogens-11-00479-t001:** Assays used for detection of Merkel cell polyomavirus in skin cancer.

First Author Year, Country	Sample Type	MCPyV Detection Assay/Target	Results	Novelty
Ungari M et al., 2021, Italy [28]	15 cases of MCC FFPE sampes	Immunohistochemical profile	CK20 (14/14), Neurofilament, (12/12), Synaptophysin (14/14); Chromogranin A (11/13), PAX5 (10/12), TDT (5/12), CK7 (1/14), TTF1(0/14)	The staining pattern of Neu-N could be used to optimize MCC diagnosis.
Prezioso C et al., 2021, Italy [22]	FFPE of skin and lymph nodes with histological diagnosis of MCC	Real-time polymerase chain reaction (qPCR) *primer and probe, targeting sT* gene MCPyV Nested PCR *different MCPyV-specific primer pairs mapping VP1 and NCCR regions of the genome*	MCPyV DNA was detected in 13/26 samples (50%), only in the primary lesions.	Data supports the “hit-and-run” hypothesis and may lead to speculation regarding MCPyV being necessary only in the initial steps of MCC oncogenesis, while further mutations drive the tumor independent from the virus.
Costa PVA et al., 2021, Brazil [23]	120 patients with histopathological exams of different cutaneous neoplasms	Two different techniques of PCR: conventional *oligonucleotides complementary to the large T-antigen (LTAg) gene* real-time PCR for detection of PyV DNA. *oligonucleotides complementary to the region called the large T-antigen of each of the PyVs JCPyV, BKPyV, WUPyV, KIPyV, MCPyV, TSPyV, HPyV6, HPyV7, HPyV9, HPyV10, HPyV12, and STLPyV.*	PyV DNA was found in 25.69% of the samples: 15% in basal cell carcinoma group, 1 5% in squamous cell carcinoma, 28.57% in melanoma, 1 5% in dermatofibrosarcoma protuberans, 13.33% in Kaposi’s sarcoma, 65% in Merkel cell carcinoma (MCC), and none in normal skin.	This study highlighted the presence of PyVs in different skin tumours.
Toptan T et al., 2020, Pittsburg, USA [31]	FFPE MCC	Differential peptide subtraction (DPS) Differential mass spectrometry (dMS) Targeted analysis *SMART sequence (5′-AAGCAGTGGTATCAACGCAGAGTAC-3′) added to the 5′ end of each dMS-identified MCPyV-*	DPS identified both viral and human biomarkers (MCPyV large T antigen, CDKN2AIP, SERPINB5, and TRIM29) that discriminate between MCPyV+ and MCPyV- MCC.	Potentially novel viral sequences can be identified in infectious tumors by DPS, a robust proteomic approach that can be employed when nucleic acid-based techniques are not feasible.
Starrett GJ et al. 2020, Bethesda, MD USA [30]	71 MCC patients FFPE sections	Deep sequencing with OncoPanel, a clinically implemented, next-generation sequencing assay targeting over 400 cancer-associated genes *Illumina libraries using a KAPA HTP library kit*	Recurrent somatic alterations common across MCC and alterations specific to each class of tumor, were associated with differences in overall survival.	High-confidence virus detection is valuable for identifying the molecular mechanisms of UV and viral oncogenesis in MCC.
Boyer M et al., 2020, France [29]	Blood samples of patients with MCC at different stages	Detection of circulating tumors cells (CTCs) using the CellSearch System and the RosetteSep-DEPArray workflow *Antibodies against surface membrane markers (EpCAM, synaptophysin, CD24, CD44, CD56 and CD45)*	CellSearch detected MCC CTCs in 26% of patients, and the R-D workflow in 42% of patients.	MCPyV DNA involved in MCC oncogenesis was detected in tumor biopsies, but not in all CTCs, suggesting that tumoral cells are heterogenous.
Motavalli Khiavi F et al., 2020, Tehran, Iran [24]	FFPE sections MCC patients 60 patients with BCC and 20 patients with SCC	Quantitative real-time PCR sequencing for mutational analysis of the MCPyV LT gene *primers/TaqMan probe to amplify a segment of MCPyV* *large T antigen*	MCPyV DNA was detected in 6 (10%) of 60 BCC (basal cell carcinoma) samples, and no viral genome was found in SCCs (squamous cell carcinoma). The median number of viral DNA copies per cell was 0.7 in 6 MCPyV-positive BCC samples.	No tumor-associated mutations were found in the LT-Ag sequence of MCPyVs from positive samples. MCPyV-positive MCC samples showed no tumor-associated mutations in the LT-Ag sequence.
Neto CF et al., 2019, Brazil [25]	MCC tumoral skin FFPE specimens non-MCC skin FFPE cancers were also analyzed.	Polymerase chain reaction (PCR) (conventional and real-time) for detection of MCPyV DNA gene region of polyoma LT MCPy primer sequences *LT.1F 5′-CCACAGCCAGAGCTCTTCCT-3′* *LT.1R 5′-TGGTGGTCTCCTCTCTGCTACTG-3′*	All MCC samples available (13) tested positive for the presence of MCPyV DNA. MCPyV DNA detection rate was higher in patients with MCC than in the other group, and its analysis was statistically significant (*p* < 0.01).	In this Brazilian cohort of patients, an association between MCPyV and MCC was proven.
Kervarrec T et al., 2018, France [26]	12 conventional MCCs and 12 cutaneous squamous cell carcinomas as controls	MCPyV viral status was obtained by combining two independent molecular procedures. *2 nested pairs of primers (LT1n, forward 5′-GGCATGCCTGTGAATTAGGA-3′ and reverse 5′-TGTAAGGGGGCTTGCATAAA-3′; and VP1n, forward 5′-TGCAAATCCAGAGGTTCTCC-3′ and reverse 5′-GCAGATGTGGGAGGCAATA-3′)*	Half of the combined MCC cases were positive for MCPyV in the neuroendocrine component.	The viral positivity in half of the combined MCC cases is indicative of similar carcinogenesis routes for combined and conventional MCC.
Álvarez-Argüelles ME et al., 2017, Spain [27]	34 FFPE MCC samples) and six non-MCC samples	MCPyV was quantified using quantitative real-time PCR (qRT-PCR) *targeted the VP1 gene from EU375803 genbank sequence of MCPyV*	In 31 (91.2%) MCC-individuals, MCPyV was detected. No virus was observed in any of the non-MCC tumors.	MCPyV was very frequent in MCC. The amplification techniques described here are suitable for detecting the presence of MCPyV virus in MCC and are easy to apply.
Wang L et al., 2017, USA [32]	87 MCCs from 75 patients	RNAscope probe targeting MCPyV T antigen transcripts on tissue microarrays (TMA) and whole-tissue sections *Hs-V-MCPyV-LT-ST-Ag*	RNA in situ hybridization (RNA-ISH) demonstrated the presence of MCPyV in 37 of 75 cases (49.3%).	RNA-ISH has a sensitivity comparable to qPCR for detecting the MCPyV and allows for correlation with tissue morphology.
Arvia R et al, 2017, Italy [33]	76 FFPE cutaneous biopsies	Two assays (qPCR and ddPCR) for MCPyV detection and quantification in formalin fixed paraffin embedded (FFPE) tissue samples *Primer Sequence (5′–3′)* *Primer Forward CCCTTTGGAGCAAATTCCA* *Primer Reverse CTGACCTCATCAAACATAGAGAA* *Probe CAAAATATCCACAAGCTCAGAAGTGA*	The number of positive samples obtained by droplet digital PCR (ddPCR) was higher than that obtained by qPCR (45% and 37%, respectively).	The ddPCR represents a better MCPyV detection method in FFPE biopsies, especially those containing low numbers of copies of the viral genome.
Paulson KG 2017, Seattle WA [21]	219 patients with newly diagnosed MCC were followed prospectively (median follow-up, 1.9 years).	MCPyV-oncoprotein antibody detection *Glutathione-S-transferase(GST)-tagged MCPyV small T-antigen*	Antibodies to MCPyV oncoproteins were rare among healthy individuals (1%), but were present in most patients with MCC [52%]; *p* < 0.01).	The clinical management of newly diagnosed MCC patients can be optimized by determining the oncoprotein antibody titer. Thus, the patients can better be stratified into a higher risk seronegative cohort, in which radiological imaging techniques may play a more prominent role, and into a lower risk seropositive cohort, in which the oncoprotein antibody titer can be used to track the disease status.

**Table 2 pathogens-11-00479-t002:** Carcinogenesis mechanisms of MCPyV in skin cancer.

First Author, Year, Country	Carcinogenesis Mechanism	Clinical Importance
Krump NA et al., 2021, Pennsylvania USA [34]	Primary human dermal fibroblasts (HDFs) can support MCPyV infection The onset of MCPyV replication and early Gene expression induces an inflammatory cytokine and interferon-stimulated gene (ISG) response.	Exploring how MCPyV interacts with innate immunity during its infectious cycle. Understanding the biology of MCPyV could lead to targeted therapies for MCPyV-associated MCC.
Guadagni S et al., 2020, Italy [35]	Identified the relationship between MCPyV and oncogenic alternative Δ exon 6–7 TrkAIII splicing in fresh, nonfixed, MCPyV-positive MCC metastasis	Identifies patients who may benefit from the following: Inhibitors of MCPyV T-antigen and/or TrkAIII expression or Clinically approved Trk kinase inhibitors: larotrectinib or entrectinib
Zhao J et al., 2020, Dallas, TX [36]	MCPyV sT-induced ncNF-κB signaling is an essential tumorigenic pathway in MCC	The first identification of the ncNF-κB signaling pathway activation by any polyomavirus and its critical role in MCC tumorigenesis.
Nwogu N et al., 2020, Pennsylvania USA [37]	MCPyV sT-mediated MMP-9 activation is driven through the large T stabilization domain (LSD)”, a known E3 ligase-targeting domain, in MCC.	Metastatic MCC may be treated in the future with a novel approach, in which MMP-9 may serve as the biochemical culprit for treatment targeting and development.
Gupta P et al., 2020, Lyon, France [38]	Twenty-eight genes were revealed to be specifically deregulated by MCPyV, using a comparison of gene expression profiles. The MCPyV early gene downregulated the expression of the N-myc downstream-regulated gene 1 (NDRG1) (a tumor supressor) in MCPyV gene-expressing NIKs and hTERT-MCPyV gene-expressing human keratinocytes (HK) compared to their expression in the controls.	New paradigms of molecular targeted therapies can provide hope for patients affected by this highly aggressive cancer.
Longino NV et al., 2019, Seattle Washington [39]	The identification of CD4+ T-cell responses against six MCPyV epitopes One epitope was of particular interest, because it included a conserved, essential viral oncogenic domain which binds to and/or disables the cellular retinoblastoma (Rb) tumor suppressor.	Therapeutic vaccines may use this key step for detoxification. More in-depth studies of MCPyV-specific CD4+ T cells may use these new tools to provide a broader insight into the cancer-specific CD4+ T-cell responses.
Wu JH et al., 2019, Houston, TX, USA [40]	MCPyV small T (sT) antigen induces the activation of the DNA damage response (DDR) pathway. The hyperphosphorylation of histone H2AX is a marker of DNA damage and was observed in MCPyV-positive MCC cells in humans.	A novel link between MCPyV sT and the DDR pathway in MCC. DDR could be quantified to evaluate radiotherapy or chemotherapy response. More attention should be given to studying the implication of the DDR pathway for the management of MCC.

**Table 3 pathogens-11-00479-t003:** Detection methods used for beta HPV types.

First Author Year, Country	Type of Samples	Detection Assay/Target	Results	Novelty
Sitarz K et al, 2021 Poland [41]	Skin biopsies from 73 patients with histopathologically confirmed BCC	PCR and reverse hybridization assay to genotype *25 types* *(HPV 5, 8, 9, 12, 14, 15, 17, 19, 20, 21, 22, 23, 24, 25, 36, 37, 38, 47, 49, 75, 76, 80, 92, 93, 96), PCR with reverse hybridization assay (RHA)* *E1 β**-HPV* *gene fragment*	Statistically significant correlation between the following: the gender and BCC type, BCC type and tumour location, BCC type and exposure to UV radiation	The presence of a single HPV 93 infection is one of the risk factors for developing infiltrating lesions.
Kopeć J et al, 2020 Poland [42]	Skin biopsies from lesions and perilesional healthy area of 118 patients with NMSC (nonmelanoma skin cancers) or precancerous lesions	PCRs with different sets of primers, PCR followed by reverse hybridization and direct sequencing of PCR amplimers *two pairs of general degenerate primers* *CP65-70 (CP65/70 and CP66/69 as external and* *internal, respectively) were used for the detection of* *EV (epidermodysplasia verruciformis)-associated HPVs*	Beta-HPVs were detected in 41% of 261 biopsies examined. The most frequently iden-tified types were HPV23, HPV24 and HPV93.	HPV5 and HPV8, consi-dered high-risk carcinogenic types, were present only in a small percentage of samples. Different methods of beta HPV detection should be used.
Galati L et al., 2020 Lyon, France [43]	Healthy skin (HS) and Actinic keratosis (AK) samples	Next-generation sequencing (NGS) Actinic keratosis (AK) arises onskin damaged by UV radiation and is the precursor lesion of cutaneous squamous cell carcinoma (cSCC) *consensus primer pair FAP (FAP59\FAP64) targeting the 5′end of the L1 ORF* *new set of degenerated FAP primers (FAPM1 primer mix)*	Identification of a large number of known β and γ HPV types was achieved. In addition, 27 putative novel β and 16 γ and 4 unclassified HPVs were isolated.	HPV types of species γ-1 (e.g., HPV4) appeared to be strongly enriched in AK versus HS.
Nguyen CV et al., 2020 Chicago, IL, USA [44]	SCCs from immunosuppressed individuals, with and without voriconazole exposure	PCR analysis for HPV DNA and compared to SCC from non-immunosuppressed patients *nested PCR utilizing FAP and PGMY-GP + primer systems*	HPV DNA was detected in all groups, regardless of the immunosuppression status (80.5%), with beta HPV being the most prevalent (64.3–78.6%).	Beta HPV types 5, 8, 14, 20, and 21 were commonly detected in voriconazole exposure-associated SCC.
Rollison DE et al, 2019. Tampa, Florida [45]	Eyebrow hairs (EBHs) and skin swabs (SSWs)	DNA belonging to 46 β-HPV and 52 γ-HPV types Viral DNA detection was performed by multiplex PCR *E7 gene for HPV types* *or the N-terminal part of the large T-antigen gene for HPyV*	Prevalence of β-HPV/γ-HPV was 92%/84% and 73%/43% in SSWs and EBHs, respect-tively, with 71%/39% of patients testing positive for β-HPV/γ-HPV in both sample types.	It is important to optimize the sensitivity of cutaneous HPV detection methods using SSWs, in parallel with the specificity of EBHs, or a combination of the two. An ongoing cohort study investigating the association between cutaneous HPV and subsequent keratinocyte carcinomas will try to determine this.

**Table 4 pathogens-11-00479-t004:** Carcinogenesis mechanisms of beta HPV types in skin cancer.

First Author, Year, Country	Carcinogenesis Mechanism	Clinical Importance
Minoni L et al, 2020, Lyon, France [46]	HPV types from the beta-3 species (which includes 3 additional HPV types: 75, 76, and HPV115) were studied for their in vitro transformation properties. HPV types 49, 75, and 76 E6 and E7 (E6/E7), but not HPV115 E6 and E7 were found to inactivate the p53 and pRb pathways efficiently and to immortalize or extend the lifespan of human foreskin keratinocytes (HFKs).	E6 and E7 from beta-3 HPV types show transforming activity. There are some similar biological properties of beta-3 HPVs that are more extensively shared with mucosal high-risk HPV16 than with beta-2 HPV38.
Viarisio D et al, 2018 Heidelberg, Germany [47]	Whole-exome sequencing showed that chronic exposure to UV radiation triggers the accumulation of a large number of UV-induced DNA mutations in K14 HPV38 E6/E7 Tg mice. The number of mutations increases proportionally with the severity of the skin lesions.	The pattern of mutations in the Tg skin lesions closely resembles that of human NMSC, with the highest mutation rate in p53 and Notch genes. These data support the idea that beta HPVs only act in the initial stages of carcinogenesis, increasing the potency of the deleterious effects of UV radiation. Beta HPV38 oncoproteins act with a “hit-and-run” mechanism in UV-induced skin carcinogenesis in mice.
Pacini L et al., 2017 Lyon, France [48]	The authors suggest that beta HPVs act as cofactors in UV-induced skin oncogenesis, by altering several cellular mechanisms activated by ultraviolet radiation. TLR9, a damage recognition receptor (DRR) of cellular stress is activated by UV radiation in primary human keratinocytes (PHKs). p53 and c-Jun, transcription factors known to be activated by UV, play key roles in TLR9 expression by UV activation. The E6 and E7 oncoproteins of beta HPV38 strongly inhibit UV-activated TLR9 expression by preventing the recruitment of p53 and c-Jun to the TLR9 promoter.	These data support the idea that beta HPV types play a role in skin carcinogenesis, by preventing the activation of specific pathways upon exposure of PHKs to UV radiation.

## Data Availability

Data are contained within the article.

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
