# Peer review of "Merkel Cell Polyoma Virus and Cutaneous Human Papillomavirus Types in Skin Cancers: Optimal Detection Assays, Pathogenic Mechanisms, and Therapeutic Vaccination"

_pathogens, 2022, doi:10.3390/pathogens11040479_

Round 1
Reviewer 1 Report
In this review the authors analyse the studies published over the past 5 years regarding detection methods and carcinogenesis mechanisms for two viruses associated with skin cancer: Merkel cell polyomavirus (MCV) and human papilloma virus (HPV) types. Although the review is relatively compact, it provides insight into both the carcinogenicity mechanisms and methods of detection, as well as availability of targeted therapy and therapeutic vaccination.
However, I have a few comments:
The main of it concerns the name of Merkel cell polyomavirus. The authors should note that the correct name for the virus is Merkel cell polyomavirus, which can be abbreviated in two ways: MCPyV or MCV. As the authors use both abbreviations alternately in the text, it would be advisable to explain it at the beginning of the manuscript. The name of the virus should be correctly used not only in the title of the manuscript but also throughout the text.
Lines - 23-24, 27, 57, 58, 164,230-231 - „Merkel cell carcinoma virus” is not correct.
Line 149 – „Merkel cell virus” is not correct.
Line 48 - „age-standardized age (ASR)” - should't be corrected to „age-standardized rate (ASR)”?
Line 100 – „ ...in case of both positive and negative MCC Merkel Cell Virus.....” is not understandable and needs to be corrected „...target therapy in both MCV positive and MCV negative cases of MCC, ...”.
Line 138 – „HR HPV” needs to be explained „high-risk HPV (HR-HPV).
Table I – Sample type second section - „diagnosis” instead of „diag-nosis”; Novelty seventh section - „ No” instead of „no”; „mutations„ not „muta-tions”; Results eleventh section - „RNA-ISH” needs to be explained „ RNA in situ hybridization (RNA-ISH)”, Results twelfth section - „ddPCR” needs to be explained „droplet digital PCR (ddPCR)”; Results thirteenth section - „oncoproteins” instead of „oncopro-teins”.
Table II – Carcinogenesis mechanisms fourth section - „LSD” needs to be explained „large T stabilization domain (LSD)”.
Author Response
Point-by-point response to the reviewers’ and editor’s comments
Title: “Merkel Cell Polyoma Virus and cutaneous Human Papilloma Virus types in skin cancers: optimal detection assays, pathogenic mechanisms, and therapeutic vaccination”
We thank the reviewer for giving us the opportunity to improve the quality of our manuscript.
Reviewer #1:
Comments and Suggestions for Authors
In this review the authors analyse the studies published over the past 5 years regarding detection methods and carcinogenesis mechanisms for two viruses associated with skin cancer: Merkel cell polyomavirus (MCV) and human papilloma virus (HPV) types. Although the review is relatively compact, it provides insight into both the carcinogenicity mechanisms and methods of detection, as well as availability of targeted therapy and therapeutic vaccination.
However, I have a few comments:
The main of it concerns the name of Merkel cell polyomavirus. The authors should note that the correct name for the virus is Merkel cell polyomavirus, which can be abbreviated in two ways: MCPyV or MCV. As the authors use both abbreviations alternately in the text, it would be advisable to explain it at the beginning of the manuscript. The name of the virus should be correctly used not only in the title of the manuscript but also throughout the text.
- Lines - 23-24, 27, 57, 58, 164,230-231 - „Merkel cell carcinoma virus” is not correct.
- Response to Reviewer: We thank the reviewer for this comment. We have used the correct name of the virus in all the manuscript as suggested (lines 23, 28, 65, 67, 81, 83, 85 – 87, 89, 92, 93, 99, 101, 102, 109, 126, 145, 179, 181, 196, 201, 203, 209, 217, 221, 244, 246, 254, 260, 262, 278, 293, 313, 322).
- Line 149 – „Merkel cell virus” is not correct.
- Response to Reviewer: We thank the reviewer for this comment. We have used the correct name of the virus as suggested (line 183).
- Line 48 - „age-standardized age (ASR)” - should't be corrected to „age-standardized rate (ASR)”?
- Response to Reviewer: We thank the reviewer for this comment. We modified as suggested (lines 53 - 56).
The estimated age-standardized incidence rates in 2020, for skin melanoma and non-melanoma skin cancer, both genders, all ages, WHO Europe, mention Switzerland, Ireland, The Netherlands on the first three places with age-standardized rate (ASR) 71.1, 63.3 and 61.5 respectively, and Romania with ASR 12.0 [2].
- Line 100 – „ ...in case of both positive and negative MCC Merkel Cell Virus....” is not understandable and needs to be corrected „...target therapy in both MCV positive and MCV negative cases of MCC, ...”.
- Response to Reviewer: We thank the reviewer for this comment. We modified as suggested (lines 122 - 127).
In 2018, the International Workshop on Merkel Cell Carcinoma Research (IWMCC) Working Group underlined some open research questions regarding this primary cu-taneous neuroendocrine carcinoma, MCC: the multidisciplinary research team (e.g., virology, pathology, oncology, dermatology) raised awareness regarding the future targeted therapy in both MCPyV positive and MCPyV negative cases of MCC, and about the optimal detection assay for this virus [15].
- Line 138 – „HR HPV” needs to be explained „high-risk HPV (HR-HPV).
- Response to Reviewer: We thank the reviewer for this comment. We modified as suggested (lines 171 - 174).
This review uses high-risk HPV (HR-HPV) types as a comparison, and the authors are confident that, if for cervical cancer, developing of 3 vaccines (bi, tetra and nonavalent) was possible, it will be feasible to create a vaccine against beta HPV types [20].
- Table I – Sample type second section - „diagnosis” instead of „diag-nosis”; Novelty seventh section - „ No” instead of „no”; „mutations„ not „muta-tions”; Results eleventh section - „RNA-ISH” needs to be explained „ RNA in situ hybridization (RNA-ISH)”, Results twelfth section - „ddPCR” needs to be explained „droplet digital PCR (ddPCR)”; Results thirteenth section - „oncoproteins” instead of „oncopro-teins”.
- Response to Reviewer: We thank the reviewer for this comment. We modified as suggested (lines 203 - 206).
Table I: Assays used for detection of Merkel cell polyomavirus in skin cancer.
First Author Year, Country |
Sample type |
MCPyV detection assay / target |
Results |
Novelty |
Ungari M et al., 2021, Italy [28] |
15 cases of MCC FFPE sampes |
Immunohistochemical profile |
CK20 (14/14), Neurofilament, (12/12), Synaptophysin (14/14); Chromogranin A (11/13), PAX5 (10/12), TDT (5/12), CK7 (1/14), TTF1(0/14) |
The staining pattern of Neu-N could be used to optimize MCC diagnosis. |
Prezioso C et al., 2021, Italy [22] |
FFPE of skin and lymph nodes with histological diagnosis of MCC |
Real-Time Polymerase Chain Reaction (qPCR) primer and probe, targeting sT gene MCPyV Nested PCR different MCPyV-specific primer pairs mapping VP1 and NCCR regions of the genome |
MCPyV DNA was detected in 13/26 samples (50%), only in the primary lesions. |
Data supports the “hit and run” hypothesis and may lead to speculation regarding MCPyV being necessary only in the initial steps of MCC oncogenesis, while further mutations drive the tumor independent from the virus. |
Costa PVA et al., 2021, Brazil [23] |
120 patients with histopathological exams of different cutaneous neoplasms |
2 different techniques of PCR: conventional oligonucleotides complementary to the large T-antigen (LTAg) gene real time PCR for detection of PyV DNA. oligonucleotides complementary to the region called the large T-antigen of each of the PyVs JCPyV, BKPyV, WUPyV, KIPyV, MCPyV, TSPyV, HPyV6, HPyV7, HPyV9, HPyV10, HPyV12, and STLPyV. |
PyV DNA was found in 25.69% of the samples: 15% in basal cell carcinoma group, 1 5% in squamous cell carcinoma, 28.57% in melanoma, 1 5% in dermatofibrosarcoma protuberans, 13.33% in Kaposi sarcoma, 65% in Merkel cell carcinoma (MCC), and none in normal skin. |
This study highlighted the presence of PyVs in different skin tumours. |
Toptan T et al., 2020, Pittsburg, USA [31] |
FFPE MCC
|
Differential peptide subtraction (DPS) Differential mass spectrometry (dMS) Targeted analysis SMART sequence (5′-AAGCAGTGGTATCAACGCAGAGTAC-3′) added to the 5′ end of each dMS-identified MCPyV- |
DPS identified both viral and human biomarkers (MCPyV large T antigen, CDKN2AIP, SERPINB5, and TRIM29) that discriminate between MCPyV+ and MCPyV- MCC |
Potentially novel viral sequences can be identified in infectious tumors by DPS, a robust proteomic approach that can be employed when nucleic acid - based techniques are not feasible. |
Starrett GJ et al. 2020, Bethesda, MD USA [30] |
71 MCC patients FFPE sections |
Deep sequencing with OncoPanel, a clinically implemented, next-generation sequencing assay targeting over 400 cancer-associated genes. Illumina libraries using a KAPA HTP library kit |
Recurrent somatic alterations common across MCC and alterations specific to each class of tumor, associated with differences in overall survival. |
High-confidence virus detection is valuable for identifying the molecular mechanisms of UV and viral oncogenesis in MCC |
Boyer M et al., 2020, France [29] |
Blood samples of patients with MCC at different stages |
Detection of circulating tumors cells (CTCs) using the CellSearch System and the RosetteSep -DEPArray workflow Antibodies against surface membrane markers (EpCAM, synaptophysin, CD24, CD44, CD56 and CD45) |
CellSearch detected MCC CTCs in 26% of patients, and the R-D workflow in 42% of patients |
MCPyV DNA involved in MCC oncogenesis was detected in tumor biopsies, but not in all CTCs, suggesting that tumoral cells are heterogenous. |
Motavalli Khiavi F et al., 2020, Tehran, Iran [24] |
FFPE sections MCC patients Sixty patients with BCC and 20 patients with SCC |
Quantitative real-time PCR Sequencing for mutational analysis of the MCPyV LT gene primers/TaqMan probe to amplify a segment of MCPyV large T antigen |
MCPyV DNA was detected in 6 (10%) of 60 BCC (basal cell carcinoma) samples, and no viral genome was found in SCCs (squamous cell carcinoma). The median number of viral DNA copies per cell was 0.7 in 6 MCPyV-positive BCC samples. |
No tumor-associated mutations in the LT-Ag sequence of MCPyVs from positive samples MCPyV-positive MCC samples showed no tumor-associated mutations in the LT-Ag sequence |
Neto CF et al., 2019, Brazil [25]
|
MCC tumoral skin FFPE specimens
non-MCC skin FFPE cancers were also analyzed. |
Polymerase chain reaction (PCR) (conventional and real-time) for detection of MCPyV DNA. gene region of polyoma LT MCPy primer sequences LT.1F 5' – CCACAGCCAGAGCTCTTCCT - 3' LT.1R 5' – TGGTGGTCTCCTCTCTGCTACTG - 3' |
All MCC samples available (13) tested positive for the presence of MCPyV DNA MCPyV DNA detection rate was higher in patients with MCC than in the other group, and its analysis was statistically significant (P < 0.01). |
In this Brazilian cohort of patients, an association between MCPyV and MCC was proven.
|
Kervarrec T et al., 2018, France [26] |
12 conventional MCCs and 12 cutaneous squamous cell carcinomas as controls |
MCPyV viral status by combining two independent molecular procedures. 2 nested pairs of primers (LT1n, forward 5′-GGCATGCCTGTGAATTAGGA-3′ and reverse 5′-TGTAAGGGGGCTTGCATAAA-3′; and VP1n, forward 5′-TGCAAATCCAGAGGTTCTCC-3′ and reverse 5′-GCAGATGTGGGAGGCAATA-3′) |
Half of the combined MCC cases were positive for MCPyV in the neuroendocrine component. |
The viral positivity in half of the combined MCC cases is indicative of similar carcinogenesis routes for combined and conventional MCC. |
Álvarez-Argüelles ME et al., 2017, Spain [27] |
34 FFPE MCC samples) and six non-MCC samples |
MCPyV was quantified using quantitative Real-Time-PCR (qRT-PCR) targeted the VP1 gene from EU375803 genbank sequence of MCPyV |
In 31 (91.2%) MCC-individuals, MCPyV was detected. No virus was observed in any of the non-MCC tumors. |
MCPyV was very frequent in MCC. The amplification techniques described here are suitable for detecting the presence of MCPyV virus in MCC and easy to apply. |
Wang L et al., 2017, USA [32] |
87 MCCs from 75 patients |
RNAscope probe targeting MCPyV T antigen transcripts on tissue microarrays (TMA) and whole-tissue sections Hs-V-MCPyV-LT-ST-Ag |
RNA in situ hybridization (RNA-ISH) demonstrated the presence of MCPyV in 37 of 75 cases (49.3%) |
RNA-ISH has a sensitivity comparable to qPCR for detecting the MCPyV and allows for correlation with tissue morphology. |
Arvia R et al, 2017, Italy [33] |
76 FFPE cutaneous biopsies |
Two assays (qPCR and ddPCR) for MCPyV detection and quantifica-tion in formalin fixed paraffin embedded (FFPE) tissue samples. Primer Sequence (5′–3′) Primer Forward CCCTTTGGAGCAAATTCCA Primer Reverse CTGACCTCATCAAACATAGAGAA Probe CAAAATATCCACAAGCTCAGAAGTGA |
The number of positive samples obtained by droplet digital PCR (ddPCR) was higher than that obtained by qPCR (45% and 37% respectively). |
The ddPCR represents a better MCPyV detection method in FFPE biopsies, especially those containing low numbers of copies of the viral genome. |
Paulson KG 2017, Seattle WA [21] |
219 patients with newly diagnosed MCC were followed prospectively (median follow-up, 1.9 years). |
MCPyV-oncoprotein antibody detection Glutathione-S-transferase (GST)-tagged MCPyV small T-antigen |
Antibodies to MCPyV oncoproteins were rare among healthy individuals (1%) but were present in most patients with MCC [52%]; P < .01). |
The clinical management of newly diagnosed MCC patients can be optimized by determining the oncoprotein antibody titer. Thus, the patients can better be stratified into a higher risk seronegative cohort, in which radiological imaging techniques may play a more prominent role, and into a lower risk seropositive cohort, in which the oncoprotein antibody titer can be used to track the disease status. |
- Table II – Carcinogenesis mechanisms fourth section - „LSD” needs to be explained „large T stabilization domain (LSD)”.
- Response to Reviewer: We thank the reviewer for this comment. We modified as suggested (lines 221 - 222).
Table II: Carcinogenesis mechanisms of MCPyV in skin cancer.
First author, Year, country |
Carcinogenesis mechanism
|
Clinical importance |
Krump NA et al., 2021, Pennsylvania USA [34] |
· primary human dermal fibroblasts (HDFs) can support MCPyV infection · the onset of MCPyV replication and early gene expression induces an inflammatory cytokine and interferon-stimulated gene (ISG) response. |
· exploring how MCPyV interacts with innate immunity during its infectious cycle. · understanding the biology of MCPyV could lead to targeted therapies for MCPyV-associated MCC
|
Guadagni S et al., 2020, Italy [35] |
· identified the relationship between MCPyV and oncogenic alternative Δ exon 6-7 TrkAIII splicing in fresh, nonfixed, MCPyV-positive MCC metastasis |
Identifies patients who may benefit from: · inhibitors of MCPyV T-antigen and/or TrkAIII expression or · clinically approved Trk kinase inhibitors: larotrectinib or entrectinib |
Zhao J et al., 2020, Dallas, TX [36] |
MCPyV sT-induced ncNF-κB signaling is an essential tumorigenic pathway in MCC |
the first identification of the ncNF-κB signaling pathway activation by any polyomavirus and its critical role in MCC tumorigenesis. |
Nwogu N et al., 2020, Pennsylvania USA [37] |
MCPyV sT-mediated MMP-9 activation is driven through the large T stabilization domain (LSD)”, a known E3 ligase-targeting domain, in MCC |
metastatic MCC may be treated in the future with a novel approach, in which MMP-9 may serve as the biochemical culprit for treatment targeting and development |
Gupta P et al., 2020, Lyon, France [38] |
· 28 genes were revealed to be specifically deregulated by MCPyV, using a comparison of gene expression profiles · the MCPyV early gene downregulated the expression of the N-myc downstream-regulated gene 1 (NDRG1) (a tumor supressor) in MCPyV gene-expressing NIKs and hTERT-MCPyV gene-expressing human keratinocytes (HK) compared to their expression in the controls. |
· new paradigms of molecular targeted therapies can provide hope for patients affected by this highly aggressive cancer.
|
Longino NV et al., 2019, Seattle Washington [39] |
· the identification of CD4+ T-cell responses against six MCPyV epitopes · one epitope was of particular interest, because it included a conserved, essential viral oncogenic domain which binds to and/or disables the cellular retinoblastoma (Rb) tumor suppressor |
· therapeutic vaccines may use this key step for detoxification. · more in-depth studies of MCPyV-specific CD4+ T cells may use these new tools to provide a broader insight into the cancer-specific CD4+ T-cell responses. |
Wu JH et al., 2019, Houston, TX, USA [40] |
· MCPyV small T (sT) antigen induces the activation of the DNA damage response (DDR) pathway. · the hyperphosphorylation of histone H2AX is a marker of DNA damage and was observed in MCPyV-positive MCC cells in humans |
· a novel link between MCPyV sT and the DDR pathway in MCC. · DDR could be quantified to evaluate radiotherapy or chemotherapy response · more attention should be given to studying the implication of the DDR pathway for the management of MCC |
Reviewer 2 Report
Unambiguously, skin cancer is an aspect of very high importance and this makes the study at least interesting. The manuscript is well written, however lacks a good organisation, needs rearrangement, main conclusions and some more molecular data to be discussed. For instance, regarding the molecular detection of these viruses, a good review paper should propose the most appropriate methods for detection. This should be added
Also, in the abstract I would suggest to add 1-2 sentences providing the main inferences of the study, not only the scope and the concept; there is enough space. Same in conclusions
In line 48 please correct age-standardized age (ASR) with the correct age-standardized rate (ASR)
Why only 34 studies were selected? Based on what criteria was this selection performed? Please add an explanation
I would expect a better description and mostly a critical evaluation of the assays presented in Tables 1 and 3. Also regarding the molecular diagnosis tools, a detailed description of the method, for example which primers or probes, what gene do they target, what is the PCR product length, etc, not just refer “real time” or “nested PCR” in Table 1
Since the authors study and analysed the two viruses, they should also present some data regarding their genetic diversity. Also basic information is missing, are they both double stranded DNA viruses? Please add this info
Finally, since it is a review study, there is no need to has this structure, with materials and methods and results
Author Response
Point-by-point response to the reviewers’ and editor’s comments
Title: “Merkel Cell Polyoma Virus and cutaneous Human Papilloma Virus types in skin cancers: optimal detection assays, pathogenic mechanisms, and therapeutic vaccination”
We thank the reviewer for giving us the opportunity to improve the quality of our manuscript.
Reviewer #2:
Comments and Suggestions for Authors
Unambiguously, skin cancer is an aspect of very high importance, and this makes the study at least interesting. The manuscript is well written, however lacks a good organisation, needs rearrangement, main conclusions, and some more molecular data to be discussed. For instance, regarding the molecular detection of these viruses, a good review paper should propose the most appropriate methods for detection. This should be added.
- Also, in the abstract I would suggest to add 1-2 sentences providing the main inferences of the study, not only the scope and the concept; there is enough space. Same in conclusions
- Response to Reviewer: We thank the reviewer for this comment. We have modified the abstract as suggested (lines 18 – 32).
Abstract: Oncogenic viruses are recognized to be involved in some cancers, based on very well-established criteria of carcinogenicity. If for cervical cancer and liver cancer the responsible viruses are well-known (e.g., HPV, HBV), in the case of skin cancer there are still many studies which are trying to identify the possible viral etiologic agents as principal co-factors in the oncogenic process. We analysed scientific literature published in the last 5 years regarding mechanisms of carcinogenicity, methods of detection, available targeted therapy and vaccination for Merkel cell polyomavirus, and beta Human Papilloma Virus types, in relation to skin cancer. This review is targeted in presenting the recent findings which are supporting the involvement of these viruses in the development of some types of skin cancers. To optimize the management of skin cancer, a health condition of a very high importance, it would be ideal that the screening of skin cancer for these two analysed viruses (MCPyV and beta HPV types) to be implemented in each regional / country cancer centres’ molecular detection diagnostic platforms, with multiplex viral capability, optimal sensitivity, and specificity, clinically validated, and if possible, at ac-ceptable prices. For confirmatory diagnosis of skin cancer, another method should be used, with a different principle, like immunohistochemistry, with specific antibodies for each virus.
- Response to Reviewer: We thank the reviewer for this comment. We have modified the conclusion as suggested (lines 311 – 319).
For this scenario to be possible in case of skin cancer, there is a need for more studies to confirm the etiologic link between MCPyV and beta HPV types and skin cancer. Both viruses have DNA genomes, thus it could be possible to develop a molecular platform for multiplex genotyping assay and even quantification of these viruses. For confirmatory diagnosis of skin cancer, another method should be used, with a different principle, like immunohistochemistry, with specific antibodies for each virus. In order that the above direction to be possible, there is a need for more studies to demonstrate the association between skin cancer and these two viruses, and of course, basic research studies, to confirm the already described carcinogenesis mechanism.
In conclusion, this review underlines the necessity of interdisciplinary collaboration in assessing skin cancers, to understand the natural history of MCPyV and beta HPV types and to correlate their carcinogenesis mechanisms with future targeted therapy and vaccination.
- In line 48 please correct age-standardized age (ASR) with the correct age-standardized rate (ASR)
- Response to Reviewer: We thank the reviewer for this comment. We modified as suggested (lines 53 - 56).
The estimated age-standardized incidence rates in 2020, for skin melanoma and non-melanoma skin cancer, both genders, all ages, WHO Europe, mention Switzerland, Ireland, The Netherlands on the first three places with age-standardized rate (ASR) 71.1, 63.3 and 61.5 respectively, and Romania with ASR 12.0 [2].
- Why only 34 studies were selected? Based on what criteria was this selection performed? Please add an explanation
- Response to Reviewer: We thank the reviewer for this comment. We added comments and explanations as suggested (lines 276 - 289).
A limitation of our study is the relatively low number of analyzed studies. We performed a systematic search of the PubMed and the EMBASE databases was carried out for all the published studies on skin cancer, Merkel cell polyomavirus and beta Human Papilloma Virus types, using the following search algorithm: skin cancer AND MCPyV/ beta HPV types AND detection assays / carcinogenesis mechanism / therapeutic vaccination. We realized a systematic analysis for studies that were published in English, from the 1st of January 2017 to the 1st of February 2022, and that described the methods of detection of these two viruses in skin cancer, their tumorigenesis mechanism in this kind of cancer and possible therapeutical vaccination approaches. One possible explanation for the relatively low number of studies found is that these viruses have only recently begun to be associated with skin cancer (especially beta HPV types). Another possible explanation is that our research investigation period included the COVID pandemic, which is known to have stopped or delayed the patients’ access to medical services, and even that some research groups had delays in their activity.
- I would expect a better description and mostly a critical evaluation of the assays presented in Tables 1 and 3. Also regarding the molecular diagnosis tools, a detailed description of the method, for example which primers or probes, what gene do they target, what is the PCR product length, etc, not just refer “real time” or “nested PCR” in Table 1
- Response to Reviewer: We thank the reviewer for this comment. We have introduced the target genes and the used primers in Table I (lines 203 - 206) and Table III (lines 233 - 235) in the column ”Detection assay / target”.
Table I: Assays used for detection of Merkel cell polyomavirus in skin cancer.
First Author Year, Country |
Sample type |
MCPyV detection assay / target |
Results |
Novelty |
Ungari M et al., 2021, Italy [28] |
15 cases of MCC FFPE sampes |
Immunohistochemical profile |
CK20 (14/14), Neurofilament, (12/12), Synaptophysin (14/14); Chromogranin A (11/13), PAX5 (10/12), TDT (5/12), CK7 (1/14), TTF1(0/14) |
The staining pattern of Neu-N could be used to optimize MCC diagnosis. |
Prezioso C et al., 2021, Italy [22] |
FFPE of skin and lymph nodes with histological diagnosis of MCC |
Real-Time Polymerase Chain Reaction (qPCR) primer and probe, targeting sT gene MCPyV Nested PCR different MCPyV-specific primer pairs mapping VP1 and NCCR regions of the genome |
MCPyV DNA was detected in 13/26 samples (50%), only in the primary lesions. |
Data supports the “hit and run” hypothesis and may lead to speculation regarding MCPyV being necessary only in the initial steps of MCC oncogenesis, while further mutations drive the tumor independent from the virus. |
Costa PVA et al., 2021, Brazil [23] |
120 patients with histopathological exams of different cutaneous neoplasms |
2 different techniques of PCR: conventional oligonucleotides complementary to the large T-antigen (LTAg) gene real time PCR for detection of PyV DNA. oligonucleotides complementary to the region called the large T-antigen of each of the PyVs JCPyV, BKPyV, WUPyV, KIPyV, MCPyV, TSPyV, HPyV6, HPyV7, HPyV9, HPyV10, HPyV12, and STLPyV. |
PyV DNA was found in 25.69% of the samples: 15% in basal cell carcinoma group, 1 5% in squamous cell carcinoma, 28.57% in melanoma, 1 5% in dermatofibrosarcoma protuberans, 13.33% in Kaposi sarcoma, 65% in Merkel cell carcinoma (MCC), and none in normal skin. |
This study highlighted the presence of PyVs in different skin tumours. |
Toptan T et al., 2020, Pittsburg, USA [31] |
FFPE MCC
|
Differential peptide subtraction (DPS) Differential mass spectrometry (dMS) Targeted analysis SMART sequence (5′-AAGCAGTGGTATCAACGCAGAGTAC-3′) added to the 5′ end of each dMS-identified MCPyV- |
DPS identified both viral and human biomarkers (MCPyV large T antigen, CDKN2AIP, SERPINB5, and TRIM29) that discriminate between MCPyV+ and MCPyV- MCC |
Potentially novel viral sequences can be identified in infectious tumors by DPS, a robust proteomic approach that can be employed when nucleic acid - based techniques are not feasible. |
Starrett GJ et al. 2020, Bethesda, MD USA [30] |
71 MCC patients FFPE sections |
Deep sequencing with OncoPanel, a clinically implemented, next-generation sequencing assay targeting over 400 cancer-associated genes. Illumina libraries using a KAPA HTP library kit |
Recurrent somatic alterations common across MCC and alterations specific to each class of tumor, associated with differences in overall survival. |
High-confidence virus detection is valuable for identifying the molecular mechanisms of UV and viral oncogenesis in MCC |
Boyer M et al., 2020, France [29] |
Blood samples of patients with MCC at different stages |
Detection of circulating tumors cells (CTCs) using the CellSearch System and the RosetteSep -DEPArray workflow Antibodies against surface membrane markers (EpCAM, synaptophysin, CD24, CD44, CD56 and CD45) |
CellSearch detected MCC CTCs in 26% of patients, and the R-D workflow in 42% of patients |
MCPyV DNA involved in MCC oncogenesis was detected in tumor biopsies, but not in all CTCs, suggesting that tumoral cells are heterogenous. |
Motavalli Khiavi F et al., 2020, Tehran, Iran [24] |
FFPE sections MCC patients Sixty patients with BCC and 20 patients with SCC |
Quantitative real-time PCR Sequencing for mutational analysis of the MCPyV LT gene primers/TaqMan probe to amplify a segment of MCPyV large T antigen |
MCPyV DNA was detected in 6 (10%) of 60 BCC (basal cell carcinoma) samples, and no viral genome was found in SCCs (squamous cell carcinoma). The median number of viral DNA copies per cell was 0.7 in 6 MCPyV-positive BCC samples. |
No tumor-associated mutations in the LT-Ag sequence of MCPyVs from positive samples MCPyV-positive MCC samples showed no tumor-associated mutations in the LT-Ag sequence |
Neto CF et al., 2019, Brazil [25]
|
MCC tumoral skin FFPE specimens
non-MCC skin FFPE cancers were also analyzed. |
Polymerase chain reaction (PCR) (conventional and real-time) for detection of MCPyV DNA. gene region of polyoma LT MCPy primer sequences LT.1F 5' – CCACAGCCAGAGCTCTTCCT - 3' LT.1R 5' – TGGTGGTCTCCTCTCTGCTACTG - 3' |
All MCC samples available (13) tested positive for the presence of MCPyV DNA MCPyV DNA detection rate was higher in patients with MCC than in the other group, and its analysis was statistically significant (P < 0.01). |
In this Brazilian cohort of patients, an association between MCPyV and MCC was proven.
|
Kervarrec T et al., 2018, France [26] |
12 conventional MCCs and 12 cutaneous squamous cell carcinomas as controls |
MCPyV viral status by combining two independent molecular procedures. 2 nested pairs of primers (LT1n, forward 5′-GGCATGCCTGTGAATTAGGA-3′ and reverse 5′-TGTAAGGGGGCTTGCATAAA-3′; and VP1n, forward 5′-TGCAAATCCAGAGGTTCTCC-3′ and reverse 5′-GCAGATGTGGGAGGCAATA-3′) |
Half of the combined MCC cases were positive for MCPyV in the neuroendocrine component. |
The viral positivity in half of the combined MCC cases is indicative of similar carcinogenesis routes for combined and conventional MCC. |
Álvarez-Argüelles ME et al., 2017, Spain [27] |
34 FFPE MCC samples) and six non-MCC samples |
MCPyV was quantified using quantitative Real-Time-PCR (qRT-PCR) targeted the VP1 gene from EU375803 genbank sequence of MCPyV |
In 31 (91.2%) MCC-individuals, MCPyV was detected. No virus was observed in any of the non-MCC tumors. |
MCPyV was very frequent in MCC. The amplification techniques described here are suitable for detecting the presence of MCPyV virus in MCC and easy to apply. |
Wang L et al., 2017, USA [32] |
87 MCCs from 75 patients |
RNAscope probe targeting MCPyV T antigen transcripts on tissue microarrays (TMA) and whole-tissue sections Hs-V-MCPyV-LT-ST-Ag |
RNA in situ hybridization (RNA-ISH) demonstrated the presence of MCPyV in 37 of 75 cases (49.3%) |
RNA-ISH has a sensitivity comparable to qPCR for detecting the MCPyV and allows for correlation with tissue morphology. |
Arvia R et al, 2017, Italy [33] |
76 FFPE cutaneous biopsies |
Two assays (qPCR and ddPCR) for MCPyV detection and quantifica-tion in formalin fixed paraffin embedded (FFPE) tissue samples. Primer Sequence (5′–3′) Primer Forward CCCTTTGGAGCAAATTCCA Primer Reverse CTGACCTCATCAAACATAGAGAA Probe CAAAATATCCACAAGCTCAGAAGTGA |
The number of positive samples obtained by droplet digital PCR (ddPCR) was higher than that obtained by qPCR (45% and 37% respectively). |
The ddPCR represents a better MCPyV detection method in FFPE biopsies, especially those containing low numbers of copies of the viral genome. |
Paulson KG 2017, Seattle WA [21] |
219 patients with newly diagnosed MCC were followed prospectively (median follow-up, 1.9 years). |
MCPyV-oncoprotein anti-body detection Glutathione-S-transferase(GST)-tagged MCPyV small T-antigen |
Antibodies to MCPyV oncoproteins were rare among healthy individuals (1%) but were present in most patients with MCC [52%]; P < .01). |
The clinical management of newly diagnosed MCC patients can be optimized by determining the oncoprotein antibody titer. Thus, the patients can better be stratified into a higher risk seronegative cohort, in which radiological imaging techniques may play a more prominent role, and into a lower risk seropositive cohort, in which the oncoprotein antibody titer can be used to track the disease status. |
Table III: Detection methods used for beta HPV types.
First Author Year, Country |
Type of samples |
Detection assay / target |
Results |
Novelty |
Sitarz K et al, 2021 Poland [41] |
Skin biopsies from 73 patients with histopathologically confirmed BCC |
PCR and reverse hybridization assay to genotype 25 types (HPV 5, 8, 9, 12, 14, 15, 17, 19, 20, 21, 22, 23, 24, 25, 36, 37, 38, 47, 49, 75, 76, 80, 92, 93, 96), PCR with reverse hybridization assay (RHA) E1 β-HPV gene fragment |
Statistically significant correlation between: the gender and BCC type, BCC type and tumour location, BCC type and exposure to UV radiation |
The presence of a single HPV 93 infection is one of the risk factors for developing infiltrating lesions |
Kopeć J et al, 2020 Poland [42] |
Skin biopsies from lesions and perilesional healthy area of 118 patients with NMSC (nonmelanoma skin cancers) or precancerous lesions |
PCRs with different sets of primers, PCR followed by reverse hybridization and direct sequencing of PCR amplimers two pairs of general degenerate primers CP65-70 (CP65/70 and CP66/69 as external and internal, respectively) were used for the detection of EV (epidermodysplasia verruciformis)-associated HPVs |
Beta-HPVs were detected in 41% of 261 biopsies examined. The most frequently iden-tified types were HPV23, HPV24 and HPV93.
|
HPV5 and HPV8, consi-dered high-risk carcinogenic types, were present only in a small percentage of samples. Different methods of beta HPV detection should be used. |
Galati L et al., 2020 Lyon, France [43] |
Healthy skin (HS) and Actinic keratosis (AK) samples |
Next generation sequencing (NGS) Actinic keratosis (AK) arises onskin damaged by UV radiation and is the precursor lesion of cutaneous squamous cell carcinoma (cSCC) consensus primer pair FAP (FAP59\FAP64) targeting the 5′end of the L1 ORF new set of degenerated FAP primers (FAPM1 primer mix) |
Identification of a large number of known β and γ HPV types. In addition, 27 putative novel β and 16 γ and 4 unclassified HPVs were isolated. |
HPV types of species γ-1 (e.g., HPV4) appeared to be strongly enriched in AK versus HS. |
Nguyen CV et al., 2020 Chicago, IL, USA [44] |
SCCs from immunosuppressed individuals, with and without voriconazole exposure. |
PCR analysis for HPV DNA and compared to SCC from non-immunosuppressed patients. nested PCR utilizing FAP and PGMY-GP + primer systems |
HPV DNA was detected in all groups, regardless of the immunosuppression status (80.5%), with beta HPV being the most prevalent (64.3-78.6%). |
Beta HPV types 5, 8, 14, 20, and 21 were commonly detected in voriconazole exposure -associated SCC. |
Rollison DE et al, 2019. Tampa, Florida [45] |
Eyebrow hairs (EBHs) and skin swabs (SSWs) |
DNA belonging to 46 β-HPV and 52 γ-HPV types Viral DNA detection was performed by multiplex PCR E7 gene for HPV types or the N-terminal part of the large T-antigen gene for HPyV
|
Prevalence of β-HPV/γ-HPV was 92%/84% and 73%/43% in SSWs and EBHs, respect-tively, with 71%/39% of patients testing positive for β-HPV/γ-HPV in both sample types. |
It is important to optimize the sensitivity of cutaneous HPV detection methods using SSWs, in parallel with the specificity of EBHs, or a combination of the two. An ongoing cohort study investigating the association between cutaneous HPV and subsequent keratinocyte carcinomas will try to determine this. |
- Response to Reviewer: We thank the reviewer for this comment. We added the critical evaluation of the assay as suggested (lines 290 - 319).
We identified different molecular assays used for both analyzed viruses (e.g., PCR, Real Time PCR, Nested PCR, NGS, multiplex PCR), which are expected to have different sensitivity, specificity and positive / negative predictive value. The studies were per-formed in just a few countries, by established researchers in the field. However, the presence of MCPyV and beta HPV types is not routinely tested, for common diagnosis, in any country.
To optimize the management of skin cancer, a health condition of a very high im-portance, it would be ideal that the screening of skin cancer for these two analyzed viruses (MCPyV and beta HPV types) to be implemented in each regional / country cancer centers’ molecular detection diagnostic platforms, with multiplex viral capability. The diagnostic platform should fulfil the criteria for optimal sensitivity and specificity (as closes as possible to 100%), clinically validated (on larger cohorts of testes patients), and if possible, at acceptable prices. This approach could be possible with the apport of health programs, by recognizing the necessity of screening for this possibly viral-induced cancer. One possible model to be followed in skin cancer screening for oncogenic viruses is the case of HPV and cervical cancer. For HPV screening very strict criteria have been established, in a guideline from 2008, for an HPV DNA test requirement: the candidate test should have a clinical sensitivity not less than 90%, a clinical specificity not less than 98% and a high interlaboratory agreement of at least 92% [53]. These strict criteria were fulfilled over the years, and recently, in 2021, well-recognized researchers proved that they were able to implement screening for HPV with big molecular platforms (e.g., COBAS 6800), with an overall sensitivity of 99,1% and a relative specificity of 99,1% [54].
For this scenario to be possible in case of skin cancer, there is a need for more studies to confirm the etiologic link between MCPyV and beta HPV types and skin cancer. Both viruses have DNA genomes, thus it could be possible to develop a molecular platform for multiplex genotyping assay and even quantification of these viruses. For confirmatory diagnosis of skin cancer, another method should be used, with a different principle, like immunohistochemistry, with specific antibodies for each virus. In order that the above direction to be possible, there is a need for more studies to demonstrate the association between skin cancer and these two viruses, and of course, basic research studies, to confirm the already described carcinogenesis mechanism.
- Since the authors study and analysed the two viruses, they should also present some data regarding their genetic diversity. Also basic information is missing, are they both double stranded DNA viruses? Please add this info.
- Response to Reviewer: We thank the reviewer for this comment. We have added the suggested data (lines 67 – 82).
Merkel cell polyomavirus
The Polyomaviridae family includes numerous small, icosahedral, non-enveloped viruses, which have a double stranded DNA genome that is approximately 5000 base pairs in length, and it is packed together with histones uptake from the host cells. These viruses have a wide range of hosts, including mammals, birds, and fish [6]. The Inter-national Committee on Taxonomy of Viruses (ICTV) currently recognizes 8 different genera of polyomaviruses (Alpha-, Beta-, Gamma, Delta-, Epsilon-, Zeta-, Eta- and The-tapolyomavirus), comprised of a total of 117 species [7]. The genetic diversity of these viruses is also very great, and a characteristic co-speciation with their hosts has been observed, which is a result of genetic recombination, as it has been observed for papil-lomaviruses [8]. Many of the viruses in this family are associated with an oncogenic capacity in animal hosts, which has been observed since the discovery of murine pol-yomavirus in the 1950s, and the Polyomaviridae family was given this name because of numerous types of tumors they can induce (poly-oma) [9].
Of human polyomaviruses, MCPyV was the first one for which evidence of a car-cinogenic potential has been observed, in a rare and aggressive form of skin cancer named Merkel cell carcinoma (MCC).
- Response to Reviewer: We thank the reviewer for this comment. We have added the suggested data (lines 128 - 135).
Cutaneous HPV types
The Papillomaviridae family is comprised of small, icosahedral, non-enveloped viruses with a double stranded DNA genome and are also characterized by a great genetic diversity and wide range of hosts, including mammals, birds, reptiles, and fish. They also have a known oncogenic potential in humans, most importantly in the development of cervical cancer, but also vulvar, vaginal, penile, and oropharyngeal cancers. The human papillomaviruses which are associated with those cancers are also called mucosal, high-risk, or alpha HPV types [16].
- Finally, since it is a review study, there is no need to has this structure, with materials and methods and results
- Response to Reviewer: We thank the reviewer for this comment. We have reorganized the manuscript as suggested, with no materials and methods and results sections.

Round 2
Reviewer 2 Report
I believe the quality of the manuscript has been deeply improved, containing now all necessary info, suitable for publication